# Anti-c-MET Fab-Grb2-Gab1 Fusion Protein-Mediated Interference of c-MET Signaling Pathway Induces Methuosis in Tumor Cells

**DOI:** 10.3390/ijms231912018

**Published:** 2022-10-10

**Authors:** Xiaoqian Dou, Qinzhi Xu, Bo Dong, Guili Xu, Niliang Qian, Cuima Yang, Hongjie Li, Liting Chen, Xin Gao, Haifeng Song

**Affiliations:** 1State Key Laboratory of Proteomics, Beijing Proteome Research Center, National Center for Protein Sciences (Beijing), Beijing Institute of Lifeomics, Beijing 102206, China; 2Beijng Immunoah Pharma Tech Co., Ltd., Beijing 100071, China

**Keywords:** bio-macromolecules, Fab fusion proteins, dual intervention, signaling pathways, methuosis, intracellular delivery

## Abstract

Bio-macromolecules have potential applications in cancer treatment due to their high selectivity and efficiency in hitting therapeutic targets. However, poor cell membrane permeability has limited their broad-spectrum application in cancer treatment. The current study developed highly internalizable anti-c-MET antibody Fab fusion proteins with intracellular epitope peptide chimera to achieve the dual intervention from the extracellular to intracellular targets in tumor therapy. In vitro experiments demonstrated that the fusion proteins could interfere with the disease-associated intracellular signaling pathways and inhibit the uncontrolled proliferation of tumor cells. Importantly, investigation of the underlying mechanism revealed that these protein chimeras could induce vacuolation in treated cells, thus interfering with the normal extension and arrangement of microtubules as well as the mitosis, leading to the induction of methuosis-mediated cell death. Furthermore, in vivo tumor models indicated that certain doses of fusion proteins could inhibit the A549 xenograft tumors in NOD SCID mice. This study thus provides new ideas for the intracellular delivery of bio-macromolecules and the dual intervention against tumor cell signaling pathways.

## 1. Introduction

Abnormal cellular signal transduction is considered one of the major driving forces behind the development of several life-threatening diseases such as cancer. Recent studies suggest the high-end potentials of small-molecule inhibitors (SMIs) as promising cancer therapeutics because of their unique abilities to alter pathogenic signaling mechanisms, thus preventing disease progression [1]. Although SMIs have shed light on different cancer mechanisms and therapeutic benefits in cancer patients, their off-target effects under certain circumstances, including insufficient specificity, selectivity, and uncontrolled systemic distribution, however, are matters worthy of further investigation. To overcome these technical hurdles, approaches such as protein-based bio-macromolecular therapeutics have shown more specificity in modulating intracellular targets [2]. Targeted delivery of protein-based bio-macromolecules to the cancer site to modulate the pathogenic signaling might be a precise and minimally toxic therapy for cancer patients [3,4]. Notably, most protein-based therapies, such as monoclonal antibody (mAb) treatment, can only be effective against extracellular or membrane targets since plasma membranes act as the barrier for the passive transportation of bio-macromolecules to the intracellular space [5,6,7]. Numerous technological approaches have been developed to facilitate the delivery of therapeutic biomolecules across the plasma membrane [8]. In this context, positively charged cell-penetrating peptides (CPPs) could be a promising vehicle to deliver their cargo proteins inside the cell [9,10,11]. However, CPPs often lack cell-type specificity, which may hinder their widespread usage in biotechnological or therapeutic applications [5,12]. An alternative approach could be to encapsulate the therapeutic antibody molecules with such a biomolecule that can be internalized into the cell via endocytosis.

Increased activities of c-MET (mesenchymal–epithelial transition protein) and its ligand hepatocyte growth factor (HGF) promote cancer cells’ proliferation and metastasis in numerous cancers. Overexpression of c-MET is associated with chemotherapy resistance and poor prognosis [13]. Therefore, the c-MET/HGF axis could be a potential target for cancer therapy. Several tyrosine kinase inhibitors (TKIs), especially those targeting the c-MET variant with alternatively spliced exon 14, have been approved by the Food and Drug Administration (FDA) for clinical application in humans. However, the development of TKI resistance in cancer cells due to long-term application demerits its survival benefits [14]. Conventional bivalent mAbs against c-MET might exhibit agonistic activities by inducing its dimerization [15,16]. Hence, neither antagonistic monovalent nor bivalent anti-c-MET mAbs show clinical benefits against cancer [17,18,19]. Amivantamab, an anti-epidermal growth factor receptor (EGFR)-c-MET bispecific antibody, was approved by the FDA in 2021 for the treatment of non-small-cell lung carcinoma (NSCLC) patients harboring insertion mutations in the EGFR exon 20. Amivantamab could attenuate the hyperactivated EGFR and c-MET signaling pathways, highlighting its potential in regulating the downstream effectors of c-MET [20].

The growth-factor receptor-bound protein-2 (Grb2) and Grb2-associated binder 1 (Gab1) play critical roles in the activation of c-MET signaling cascades. A 16-amino-acid (aa) motif within the c-MET-binding domain (MBD) of Gab1 is required for its interaction with phosphorylated c-MET at Y1349 [21]. Conversely, phosphorylation of c-MET at Y1356 is required for the docking of the SH2 domain of Grb2, subsequently leading to the recruitment of Gab1 to form the c-MET–Grb2–Gab1 ternary complex. The Grb2 SH2-domain also binds the phospho-tyrosine-containing motifs of EGFR family proteins to mediate the activation of the EGFR signaling pathway [22,23]. Furthermore, the Grb2 SH3 domains bind the son of sevenless (SOS) protein, leading to the activation of Ras and mitogen-activated protein (MAP) kinase-associated signaling [24]. Therefore, the Grb2-SH2 domain and Gab1-MBD motif might function as epitope peptides to block the interaction of endogenous Grb2 or Gab1 with c-MET, thereby limiting the proliferation of cancer cells [25,26,27]. Herein, we developed a protein chimera with a potentially high affinity for the c-MET binding site on Grb2 and Gab1 (termed ZWB90-3). Efficient cellular internalization of ZWB90-3 was achieved by conjugating the Fab form of emibetuzumab targeting c-MET. We found that ZWB90-3 had a relatively higher inhibitory effect on c-MET than that on Emi-IgG4. Moreover, we assessed other anticancer properties of ZWB90-3 using both in vitro and in vivo models.

## 2. Results

### 2.1. Emi-Based Fab Acts as an Intracellular Delivery Carrier

Emibetuzumab (Emi-IgG4), a c-Met antagonist, has been reported to induce c-Met internalization from the cell surface. Plasmid DNA carrying Emi-IgG4 expression cassette as emibetuzumab analog was constructed and expressed in mammalian cells. Fab fragment (termed Emi-Fab) of Emi-IgG4 was generated by papain-mediated cleavage of Emi-IgG4. Consistent with previous reports about the depletion of cell surface c-MET [28], we validated these results by measuring the normalized median fluorescence intensity (MFI) from flow cytometry. Emi-IgG4 was observed to cause substantial depletion of c-MET on the cell surface. Conversely, Emi-Fab exhibited only partial depletion of c-MET at the same molar concentrations (Figure 1A). Moreover, confocal microscopy showed Emi-Fab-induced c-MET internalization in HCC827 cells, and Emi-IgG4 was used as the reference group (Figure 1B).

We hypothesized that Emi-Fab might have an enhanced endosomal escape activity compared to Emi-IgG4. To test this hypothesis, an intracellular split green fluorescent protein (GFP) reporter system was established in HCC827 cells [29,30]. GFP11 peptide escaping from endosomes could restore the fluorescence of cytosolic, non-fluorescent GFP1-10 protein (Appendix A). As shown in Figure 1C, compared to Emi-IgG4, Emi-Fab had an enhanced fluorescence at 6 h and 48 h detection, indicating that the endosomal escaping ability of Emi-Fab was better than Emi-IgG4. Flow cytometric analysis was consistent with the results of the confocal microscopy assay (Figure 1D). This result further supported our hypothesis of the rationality of choosing Emi-Fab as a potential intracellular cargo protein delivery system.

### 2.2. Construction, Expression, and Identification of the Recombinant Fusion Proteins

The schematic representation of Emi-Fab-based fusion proteins is shown in Figure 2A. Fusion proteins were stabilized by the interaction between the Emi-VH and Emi-VL domain and the hinge region. The knob-into-hole mutation was introduced into the CH1 region and kappa light chain to decrease the potential for mismatch of homodimers [31]. ZWB90-1 (Emi-Fab-Gab1) and ZWB90-3 (Emi-Fab-Gab1 and Grb2) were successfully purified by the Protein L affinity column; ZWB90-2 (Emi-Fab-Grb2) was excluded from this study due to the low purity and output. Sodium dodecyl sulfate-polyacrylamide gel electrophoresis (SDS-PAGE) and size-exclusion chromatography (SEC) confirmed the predicted molecular weight and purity of each protein (Figure 2B,C).

ForteBio Octet was used to detect the affinity of fusion proteins to antigen c-MET. The K_D_ value of ZWB90-1 and ZWB90-3 were equal to Emi-IgG4 if normalized to the same number of epitope of c-MET (1 nM vs. 0.65 nM) (Figure 2D). These data indicated that the fusion peptides at the C-terminal had little space effects on the Emi-Fab domain. The avidity of ZWB90-3 and Emi-IgG4 were compared in c-MET-positive cancer cells by flow cytometry, as shown in Figure 2E,F. In c-MET high expression HCC827 cells, there was little difference between ZWB90-3 and Emi-IgG4, while in A549 cells, which have low expression levels of c-MET, the EC_50_ was 719.9 nM vs. 29.42 nM, respectively.

In order to test the internalization of ZWB90-3, HCC827 cells were incubated with ZWB90-3 and assayed by a laser-scanning confocal microscope. The three-dimensional (3D) stereograms showed that ZWB90-3 distributed among the cytoplasm, indicating that the Emi-Fab-Gab1 and Grb2 fusion protein kept the intracellular delivery activity (Appendix A).

### 2.3. ZWB90-3 Inhibited Tumor Cell Proliferation and Blocked the c-MET Signaling Pathway

To measure the effect of antibodies on tumor cell proliferation, Incucyte was applied to monitor the cell status. Only a 1 µM dose of Emi-IgG4 displayed an inhibition effect on HCC827 cell proliferation. Compared to Emi-IgG4 treatment, ZWB90-1 and ZWB90-3 induced extensive, dose-dependent inhibition effects on the proliferation of HCC827 (high c-MET expression), A549 (middle c-MET expression), MBA-MD-231 (moderate c-MET expression), and MCF7 cells (very low c-MET expression), in which ZWB90-3 led to a more significant inhibition effect than ZWB90-1 (Appendix A). ZWB90-3 inhibited HCC827 and A549 proliferation with IC_50_ values of 0.492 and 1.015 µM, respectively (Figure 3A,B).

The HGF/c-Met axis plays a critical role in cellular proliferation by triggering downstream signaling pathways. We measured the ability of ZWB90-3 to block the cascade signal pathway. As shown in Figure 3C,D, compared with the control group, phosphorylation of c-MET induced by Emi-IgG4 was observed in both HCC827 and A549 cells. However, ZWB90-3 treatment caused a reduction in phospho-c-MET levels, accompanied by decreased phosphorylation of Gab1, AKT, and SHP2 levels. The quantitation of fold change in the relative phosphorylation level of each target protein in two different cell lines is presented in Appendix A. Of those, Akt phosphorylation was remarkedly reduced in HCC827 cells after ZWB90-3 treatment. Other phosphorylated proteins in HCC827 cells or all phosphorylated proteins in A549 cells showed a downregulation trend, but the performance was weak. The Emi-IgG4 upregulation effect might have been due to bivalent Emi-IgG4-mediated dimerization and autophosphorylation of c-MET. Moreover, these data suggest that ZWB90-3 efficiently interferes with intracellular adaptor proteins Gab1 and Grb2 and inhibits its downstream signaling.

### 2.4. ZWB90-3 Induced Cell Cycle Arrest at the G2/M-Phase

To explore the mechanism of ZWB90-3 on the cell growth arrest of HCC827 and A549 cells, cell cycle progression in both cell lines was detected by flow cytometry. As shown in Figure 4A, after ZWB90-3 treatment, the cell cycle of HCC827 was arrested in the G2/M phase compared with the control (PBS-treated) and Emi-IgG4. The percentage of cells arrested in the G2/M phase increased in a time- and dose-dependent manner (Figure 4B). The cell cycle of A549 treated with 3 μM ZWB90-3 was also arrested in the G2/M phase (Appendix A).

The IncuCyte^®^ imaging system was employed to monitor the cellular morphology of HCC827 cells treated with ZWB90-3, which revealed critical stages of the cell cycle (cell cycle arrest) and morphological alterations (cell size increase) in response to drug treatment (Appendix A) compared with mock-treated control cells (Appendix A).

Abnormal occurrence in microtubule polymerization leads to mitotic failure; the morphology of the microtubule skeleton was visualized by immunofluorescent staining of α-tubulin [32]. As shown in Figure 4C, ZWB90-3 affected microtubule organization, the microtubule skeleton was absent from the nuclear periphery, and the weak fluorescence intensity of α-tubulin was consistent with the result of Western blotting (Figure 4D). These results suggested that ZWB90-3 may interfere with the extension and arrangement of microtubules, thereby inducing G2/M phase arrest in HCC827 cells.

### 2.5. ZWB90-3 Induced Methuosis of Tumor Cells

To intensively understand the morphology changes caused by ZWB90-3, long-term incubation was performed. As shown in Figure 5A, a large amount of vacuole accumulation in the cytoplasm was observed in HCC827 and A549 cells treated with ZWB90-3 for 72 h, which is the hallmark of methuosis, a novel non-apoptotic cell death. Emi-IgG4 treatment did not induce the same effects. Previous studies showed that methuosis could be triggered by alterations in the trafficking of the clathrin-independent endosome; endosome tracers were applied to verify our hypothesis [33]. On the basis of confocal microscopy, we found that a portion of vacuoles was co-localized with early endosome marker Rab5, late endocytic membrane marker Rab7, and lysosomal-associated membrane protein 1 (LAMP1), and the vacuoles did not co-localize with the lysosomal marker LysoTracker (Figure 5B).

Methuosis was initially identified in glioblastoma cells with ectopic expression of activated Ras. Molecule-induced methuosis has been reported in a broad spectrum of tumor cells [34,35]. ZWB90-3 caused a striking cytoplasmic vacuolization, which might have been due to the interference by the Grb2-SH2 domain of ZWB90-3 through competitive binding. Since there was no Grb2-SH3 domain to interact with SOS, Ras activation could not be controlled in a negative feedback loop. In line with our expectations, the Ras level was upregulated (Appendix A).

### 2.6. ZWB90-3 Inhibited Tumor Growth In Vivo

The anti-tumor activity of ZWB90-3 was observed in the A549 xenografted NSG female mice modeling NSCLC. The drug administration scheme is presented in Figure 6A. The results showed that the dose of 25 mg/kg of ZWB90-3 treatment for one week significantly inhibited tumor growth compared to the vehicle group (*p* < 0.05), while the 10 mg/kg of ZWB90-3 treatment showed a lower tumor inhibitory effect. Emi-IgG4 (30 mg/kg) treatment had a small inhibitory effect (Figure 6B). Pictures of the tumors were taken (Figure 6D). The reduction in tumor growth treated with 25 mg/kg ZWB90-3 was coincident with the data on tumor mass and body weight ratio (Figure 6E). There were no significant differences in the body weights among the groups, indicating limited system toxicity (Figure 6C).

Although the experimental animal models were NSG mice, which were deficient in T-, B-, and NK-cells, representing severe combined immunodeficiency (SCID), a low level of immunoreactivity was observed in the ZWB90-3 group, especially in the spleen. The spleen sizes and spleen-to-body weight ratios showed significant differences in the ZWB90-3 high-concentration group compared to the Emi-IgG4 group (*p* < 0.05) (Appendix A). Phenoptics quantitative pathology research has been widely used for the quantification of cancer-immune interactions, which helps to reveal mechanisms of cancer immunotherapy. To address the effect of methuosis in A549 cells on the immune system, the tumor mass was subjected to a multiplex immunofluorescence (IF) workflow (PerkinElmer, Waltham, MA, USA). Multiplex IF analysis of the tumor tissues regarding the infiltration of monocyte (CD11B+), neutrophils (CD11B+, Ly6G+), and macrophages (CD11B+, F4/80+) showed significant increase in ZWB90-3 groups (Appendix A). Representative images are shown in Appendix A.

## 3. Discussion

Numerous studies have implicated the importance of delivering bio-macromolecules into cells to interfere with the intracellular signaling pathway for cancer therapy. The biggest challenges in realizing intracellular macromolecule delivery via the endocytic pathway are the efficient cellular uptake and the endosomal escaping ability to avoid degradation by specific lysosomal enzymes [36,37]. Moreover, c-MET has emerged as a key target for modulating the c-MET/HGF axis for tumor therapy, showing promising therapeutic potential for c-MET-positive tumors. However, c-MET as a therapeutic target has several technical limitations, such as the agonistic rather than antagonistic properties of some bivalent c-MET antibodies [38]. Our study characterized the properties of antibody fusion proteins using the internalization ability of emibetuzumab Fab to achieve intracellular peptide delivery, which showed no agonistic activity of c-MET dimerization.

We demonstrated that Fab’s retention of endocytic capacity and the inhibition of c-MET degradation could be enhanced by its endosomal escaping capacity. The split GFP reporter system further verified the endosomal escaping ability of Emi-Fab. The mechanism of the strong recycling capacity of Emi-Fab might be related to a lack of the Fc portion of a full antibody for binding to the TRIM21 protein. TRIM21′s interaction with immunoglobulins was found to occur with specificity and extremely high affinity through binding to residues on Fc. TRIM21 is a member of the tripartite motif (TRIM) protein family of RING E3 ubiquitin ligases. The TRIM21/Fc interaction mediates proteasomal degradation [39]. Hence, we speculated that Emi-Fab could avoid interaction with TRIM21 to reduce protein degradation. However, more validation studies are needed to be performed.

Intracellular delivery peptides as dominant-negative mutants could block the binding of endogenous adaptor proteins to their corresponding binding partners. The results of subsequent studies suggest the effectiveness of dual interventions from extracellular to intracellular targets over extracellular ones in tumor therapy. ZWB90-3 exhibited significant interference with cell signaling pathways, thus improving the tumor-suppression efficacy in different tumor cell lines, including HCC827, A549, and MBA-MD-231 cells. The Grb2-SH2 domain has been reported to inhibit cell growth by suppressing the Grb2-Ras pathway in vitro [40]. It has been demonstrated that direct binding of Grb2 activated SHP2 phosphatase in the absence of receptor tyrosine kinase upregulation [41]. In this study, the effects of ZWB90-3 on multiple signaling pathways as well as on the downregulation of phosphorylation of c-MET, Gab1, AKT, and SHP2 were investigated. This proved that the part of ZWB90-3 targeting intracellular targets interfered with the intracellular signaling pathway and could downregulate the phosphorylation of Gab1 and downstream signaling molecules from Gab1. The current study did not consider HGF, but rather mainly focused on intracellular signaling and achieved intracellular target intervention.

Another key property of ZWB90-3 is its ability to cause the formation of intracellular vacuoles, which interfere with the normal extension and arrangement of microtubules, thereby inducing G2-phase arrest and subsequent inhibition of the proliferation of tumor cells. The excessive accumulation of vacuoles can induce methuosis. Similarly, as a macromolecule, CD99 activation triggered by mAb can cause the formation of the IGF-1R/Ras/Rac1 complex, which accumulates in vacuoles and causes methuosis but does not induce apoptosis [42]. The larger vacuoles can acquire endosomal characteristics but do not merge with lysosomal compartments. Finally, persistently accumulating vacuoles may take over the cytoplasmic space, leading to the loss of metabolic activities and rupture of cell membranes. The mechanisms might be dependent on the overexpression of Ras and is associated with the changes in the regulation of actin dynamics. The protein levels of active Ras signaling components reach a threshold to stimulate Rac1 and inactivate Arf6 (a small GTPase), thereby disturbing the recycling of clathrin-dependent endosomes to plasma membranes. Hence, the intracellular vacuoles fail to recycle to the cell membranes or fuse with lysosomes [35,43]. Our study has shown the efficiency of ZWB90-3 in inducing methuosis. Similarly, in our study, ZWB90-3 induced the upregulation of intracellular Ras. As a result, ZWB90-3 affected endosome–lysosome fusion and inhibited acidification as well as protein degradation in the lysosomes of cultured cells.

This study demonstrated that administration of 25 mg/kg of ZWB90-3 for one week could inhibit the A549-tumor-cell-line-derived xenograft tumors in mice. On the other hand, the 10 mg/kg dose showed a minimal tumor inhibitory effect. This might be attributed to the fact that either the A549 tumor models might not be an appropriate choice to reflect the medicinal efficacy of ZWB90-3, or that the Fab fragment peptide was unstable with a short half-life in vivo. In addition, only treatment with ZWB90-3 resulted in an enlarged spleen phenotype and inflammatory cell infiltration into the tumor. We speculate that methuosis observed in vitro may be a mode of immunogenic cell death (ICD), which could stimulate a robust immune response against dead-cell antigens [44,45], which warrants further investigations.

In summary, these data provided insights that emibetuzumab Fab could achieve peptide-mediated intracellular delivery of therapeutic cargo molecules, providing certain levels of tumor-suppressing effects both in vitro and in vivo. This study has major implications for the development of therapeutic bio-macromolecules to efficiently modulate intracellular targets for cancer therapy. Furthermore, the dual intervention of extracellular and intracellular targets would be a novel idea in tumor treatment.

## 4. Materials and Methods

### 4.1. Cell Culture and Lentiviral Transfection

HCC827, MBA-MD-231, and MCF7 cell lines were purchased from the Research Facilities of Peking Union Medical College (PUMC) Cell Bank (Beijing, China). The A549 cell line was purchased from Shanghai Biowing Biotechnology Co (Shanghai, China). A549 and MCF7 cells were cultured in Dulbecco’s modified Eagle’s medium (DMEM) (Corning, NY, USA) supplied with 10% fetal bovine serum (FBS) (Gemini Bio, CA, USA). HCC827 and MBA-MD-231 cells were maintained in RPMI 1640 medium (Corning, NY, USA) supplied with 10% FBS. 293F cells were grown in Freestyle™ 293 expression medium (OPM, Shanghai, China). All cells were maintained at 37 °C in a humidified chamber containing 5% CO_2_.

Lentiviral particles were produced to transfect GFP1-10 (#80409, Addgene, Watertown, MA, USA) gene into HCC827 cells according to the protocol of Lentivirus Production (Addgene) [46]. GFP1-10-positive cells were selected with puromycin (3 μg/mL).

### 4.2. Protein Expression and Purification

Protein sequences of the emibetuzumab complementarity-determining region (CDR) (disclosed in WO 2010/059654) [28], Gab1-MBD16, and Grb2-SH2 domain are described in Appendix A. Antibodies used in this study were designed on the basis of the X-FAB platform (patent no. CN110669137B). DNA sequences encoding the light and heavy chains of target proteins were synthesized and cloned into the pQK expression vector by General Inc. All the constructed vectors were sequence-validated by the Sanger sequencing method (BioMed, Beijing, China).

Plasmid DNA carrying the expression cassettes for the paired heavy and light chains were transiently co-transfected into 293F cells [47]. Emibetuzumab-IgG4 (Emi-IgG4) antibody and Fab-based (Emi-Fab) proteins were purified from the supernatant of the 293Fv cells using the Protein A and Protein L (GE Healthcare, WI, USA) affinity column chromatography, respectively. Purified proteins were dialyzed against phosphate-buffered saline (PBS) and stored at −80 °C until further analysis. Protein concentration was determined using a DropSense16-Micro-Volume Spectrophotometer (Unchained Labs, CA, USA). No city

### 4.3. Flow Cytometry Analysis

For the detection of cell surface c-MET, cells were plated in duplicates into 6-well plates and treated with either Emi-IgG4 or Emi-Fab. Cells were subsequently harvested using enzyme-free dissociation solution (Gibco, Life Technologies, Carlsbad, CA, USA) at different time points and labeled with FITC-anti-c-MET antibody (Thermo Fisher Scientific, MA, USA), which recognized a separate MET epitope from Emi-IgG4, at 4°C for 30 min, and the cells expressing membrane c-MET were quantified using flow cytometry (Agilent Technologies, Santa Clara, CA, USA). For detecting the GFP expression in HCC827-GFP1-10 cells, cells were cultured overnight and incubated with either Emi-Fab-GFP11 (3 µM) or Emi-IgG4-GFP11 (1.5 µM) for 6 h or 44 h, respectively. Cells were then harvested and analyzed using a flow cytometer to determine the GFP-expressing cell count.

For detecting the avidity of fusion proteins, cells were harvested and resuspended in PBS with 2% FBS to obtain a density of 2 × 10^6^ cells/mL. ZWB90-3 and Emi-IgG4 were diluted to the maximum concentration of 10 µM with a threefold gradient dilution. Then, 50 µL of prediluted antibodies were added to 50 µL of cell suspensions, and the mixture was incubated at 4 °C for 30 min. APC anti-human light chain kappa (Biolegend, San Diego, CA, USA) was used as the secondary antibody. Flow cytometry results were expressed as median fluorescence intensity and fitted curve using GraphPad Prism.

For cell cycle analysis, tumor cells were treated with either Emi-IgG4 or ZWB90-3, harvested, and fixed in 70% (*v*/*v*) cold ethanol at −20 °C overnight. The fixed cells were stained with RNase-containing propidium iodide (PI) solution (Sigma-Aldrich), and the DNA contents were analyzed using a flow cytometer.

### 4.4. Cell Proliferation Assay

Tumor cells were plated in duplicates in 96-well plates and labeled with Incucyte^®^ Nuclight Rapid Red dye (Sartorius, Goettingen, Germany) for live cell imaging and real-time cell quantitation of cell proliferation rate. The Incucyte live-cell imaging system (Sartorius, Goettingen, Germany) was used to assess cells’ status at 3 h intervals. The half-maximal inhibitory concentration (IC_50_) values were calculated by the Incucyte^®^ live-cell Analysis system. The cellular inhibition ratio was calculated using Equation (1).
Inhibition rate = (control group − experiment group)/control group × 100%(1)

### 4.5. Confocal Immunofluorescence (IF) Microscopy

The internalization of antibodies (Emi-IgG4 and Emi-Fab) and fusion proteins (ZWB90-3) across cell membranes was detected by confocal microscopy. Cells were grown on 12 mm coverslips in 24-well culture plates. Emi-IgG4, Emi-Fab, and ZWB90-3 were labeled with PE fluorescent dye (Abcam, Waltham, MA, USA) and co-incubated at 50 nM with cells for 2 h at 37 °C. Then, cell membranes were counterstained with fluorescent PKH67 (Sigma-Aldrich, St. Louis, MO, USA). Cells were then fixed with 3.7% formaldehyde for 15 min at room temperature, and the nuclei were stained with a 4’,6-diamidino-2-phenylindole (DAPI) reagent (Sigma-Aldrich).

The vacuoles were labeled with markers as described elsewhere [43]. HCC827 cells were incubated with 3 μM of ZWB90-3 for 60 h. LysoTracker Red DND-99 (Thermo Fisher Scientific, Waltham, MA, USA) was applied as the indicator of intracellular acidic compartments/lysosomes. For staining with anti-Rab5 and anti-Rab7 antibodies, HCC827 cells were fixed with ice-cold methanol and incubated with the indicated primary antibodies, and FITC anti-rabbit IgG was used as the secondary antibody (Jackson ImmunoResearch, MD, USA). Lysosomal-associated membrane protein 1 (LAMP-1), a transmembrane glycoprotein, was detected using FITC-LAMP-1 (Thermo Fisher Scientific, MA, USA). The PE-α-tubulin (Santa Cruz, Dallas, TX, USA) staining was performed as described above. Cell nuclei were stained with Hoechst 33342. Confocal images were captured by a Nikon AX confocal laser microscope (Nikon, Tokyo, Japan).

### 4.6. Western Blotting (WB)

For WB analysis, A549 and HCC827 cells were incubated with ZWB90-3 (3 µM and 0.3 µM, respectively) or Emi-IgG4 (1.5 µM and 0.15 µM, respectively) for 72 h. Whole-cell lysates were prepared in radio-immunoprecipitation assay (RIPA) buffer (NCM, Suzhou, China) with a 1% protease and phosphatase inhibitors cocktail (NCM, Suzhou, China). Protein concentrations of cell lysates were quantified using a bicinchoninic acid protein (BCA) assay kit (Tiangen, China). The antibodies for c-MET, phospho(p)-c-MET, Gab1, p-Gab1, AKT, p-AKT, SHP2, p-SHP2, ERK1/2, p-ERK1/2, RAS, GAPDH, and β-actin were obtained from Cell Signaling Technology (CST, MA, USA), and the anti-α-tubulin antibody was purchased from Santa Cruz Biotechnology (CA, USA). Enhanced chemiluminescence (ECL) detection reagents (NCM, Suzhou, China) were used to develop the protein signals (Clinx Science Instruments, Shanghai, China).

### 4.7. Xenograft Mouse Models

All animal studies were conducted under the guidelines of the Institutional Animal Care and Use Committee (IACUC) at Beijing Immunoah Pharma Tech Co. (Beijing, China). The NOD SCID gamma (NSG) mice were obtained from Shanghai Model Organisms (Shanghai, China). A total of 5 × 10^6^ A549 cells were injected subcutaneously into the right flank of NSG mice in 0.1 mL PBS. The mice were randomly divided into four groups when the tumor size reached a predetermined volume of 60 to 80 mm^3^. Drugs were administered by injection through the tail vein. ZWB90-3 was administered daily for one week, while Emi-IgG4 was administered three times a week. The tumor sizes and body weights of the mice were measured twice a week for 29 days. Tumor volume was calculated using Equation (2).
V (mm^3^) = longest diameter × shortest diameter × shortest diameter/2(2)

At the endpoint of the experiment, the mice were sacrificed by CO_2_ inhalation, and the tumor and spleen tissues were harvested, weighed, and photographed.

### 4.8. Statistical Analysis

All the data were analyzed using GraphPad Prism 8.0.2 (GraphPad Software, Inc., San Diego, CA, USA) and represented as mean ± SEM unless otherwise noted. Statistical analyses were performed using Statistical Product and Service Solutions (SPSS) software (IBM, Chicago, IL, USA). Significance was marked as follows: * *p* < 0.05; ** *p* < 0.01, or *** *p* < 0.001.

## Figures and Tables

**Figure 1 ijms-23-12018-f001:**
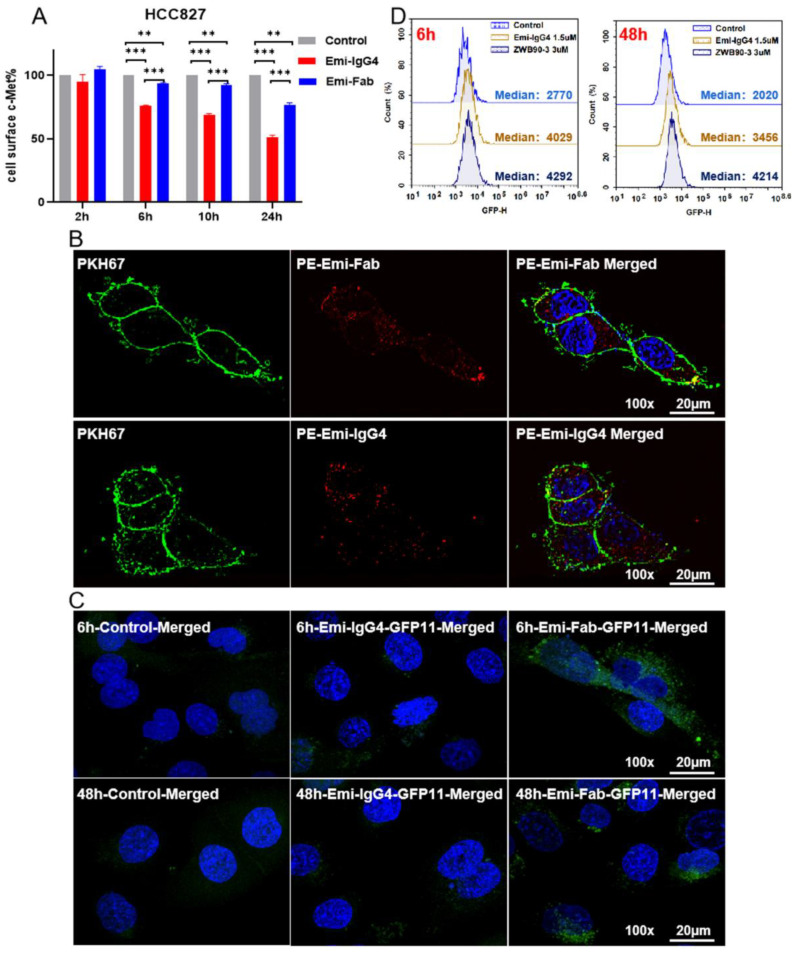
Function, internalization, and endosome-escaping ability of Emi-Fab. (**A**) Effects of Emi-Fab and Emi-IgG4 on c-MET degradation. Results are shown as the mean ± SD and are pooled from two independent experiments. ** *p* < 0.01 and *** *p* < 0.001. (**B**) Internalization of Emi-IgG4 and Emi-Fab at equal molar concentrations. Images were acquired at 100× magnification. Scale bars: 20 μm. (**C**) Representative microscopy images of cytosolic fluorescence distribution in HCC827-GFP1-10 cells treated with Emi-IgG4-GFP11 (250 nM) or Emi-Fab (500 nM) for 6 h and 48 h, respectively, as indicated. (**D**) Histograms of GFPβ1-10 expressing HCC827 cells treated with Emi-Fab-GFPβ11 or Emi-IgG4-GFPβ11 as measured by flow cytometry.

**Figure 2 ijms-23-12018-f002:**
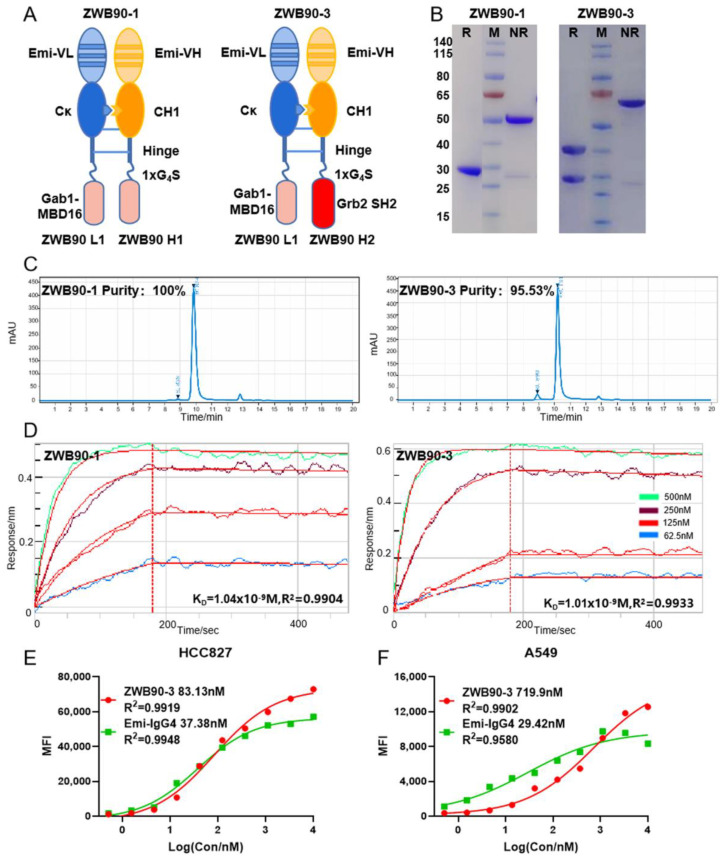
Generation and identification of ZWB90-1 and ZWB90-3. (**A**) Schematic representation of the design of ZWB90-1 and ZWB90-3. (**B**) Purities and molecular weights of ZWB90-1 and ZWB90-3 were detected by staining with Coomassie brilliant blue. Data showed expected band sizes and satisfactory purity levels. (**C**) The purities of ZWB90-1 and ZWB90-3 were also confirmed using the SEC. (**D**) ForteBio Octet determination of the affinities of ZWB90-1 and ZWB90-3 toward c-MET. The binding affinity parameter K_D_ was calculated, as reported by ForteBio Data Analysis Software 8.0 (Fremont, CA, USA). (**E**,**F**) Dose-dependent binding of ZWB90-3 and Emi-IgG4 in HCC827 cells and A549 cells. The avidities were calculated using a four-parameter logistic curve analysis with GraphPad software (San Diego, CA, USA). R^2^ is the coefficient of determination for estimating the goodness of the curve fit.

**Figure 3 ijms-23-12018-f003:**
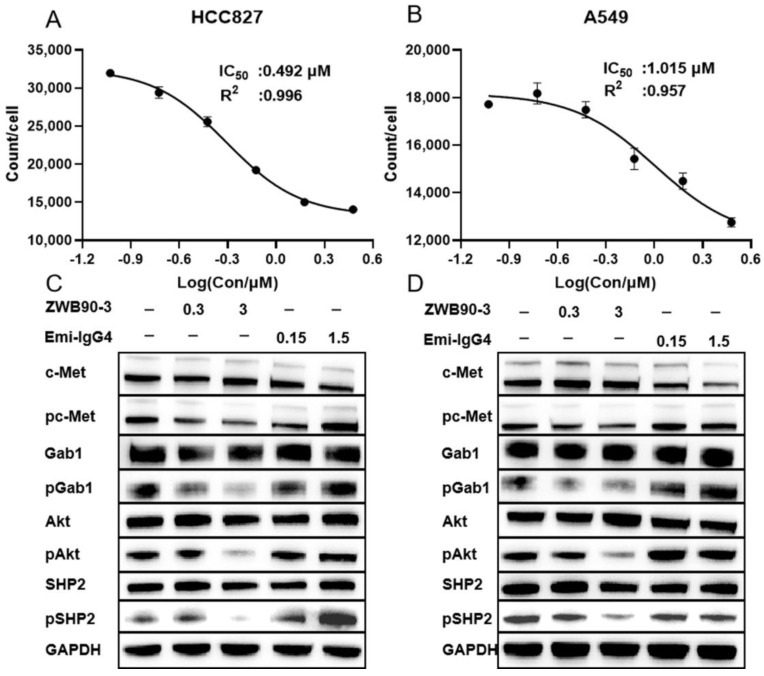
Effects of ZWB90-3 on the inhibition of cell proliferation and downstream signaling. (**A**,**B**) The IC_50_ values were calculated using a four-parameter logistic curve in HCC827 and A549 cells, respectively. (**C**,**D**) Western blot (WB) analysis of c-MET, p-c-MET, Gab1, p-Gab1, AKT, p-AKT, SHP2, and p-SHP2 expression levels in HCC827 and A549 cells, respectively.

**Figure 4 ijms-23-12018-f004:**
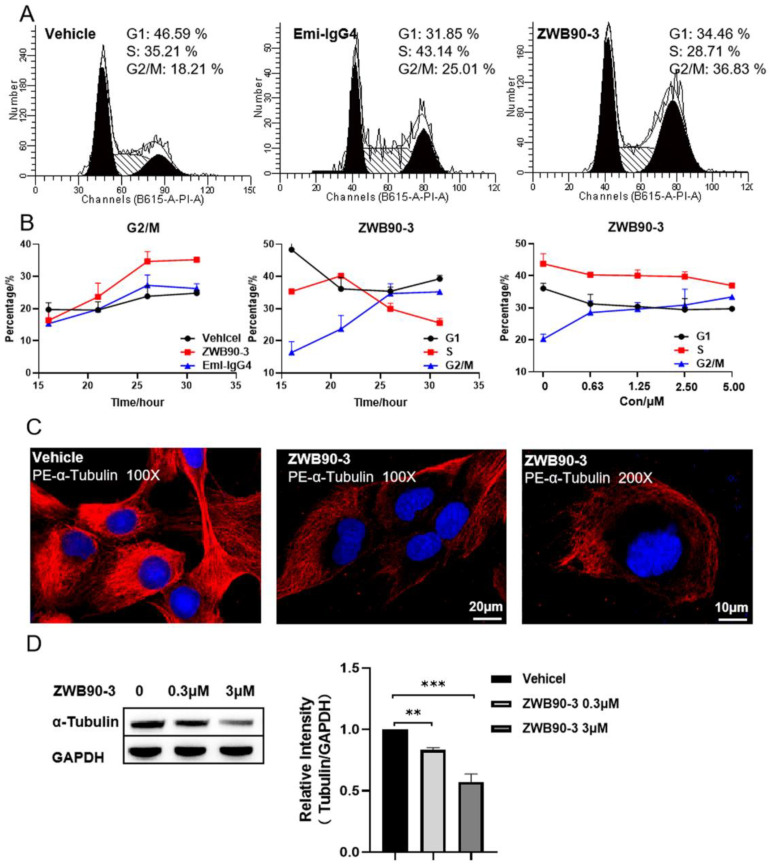
ZWB90-3-induced G2/M-phase arrest and interference in the microtubule arrangement. (**A**) HCC827 cells were treated with PBS (vehicle) or ZWB90-3 or Emi-IgG4 for 26 h, and total DNA contents were analyzed using flow cytometry. Numbers within the boxes are proportions to the G0/G1, S, and G2/M phases. (**B**) The curve of the change over time between groups in the G2/M phase (**left**). The time-course curve of the ZWB90-3-induced G2/M-phase arrest (**middle**). The dose–response curve of ZWB90-3-induced G2/M-phase arrest (**right**). (**C**) Immunofluorescence (IF) staining of α-tubulin in the vehicle and ZWB90-3-induced cells. Images were acquired at 100× or 200× magnification. Scale bars: 20 μm or 10 μm. (**D**) Western blot images, where α-tubulin and GAPDH served as the loading controls. ** *p* < 0.01 and *** *p* < 0.001 were considered significant as compared to the vehicle control.

**Figure 5 ijms-23-12018-f005:**
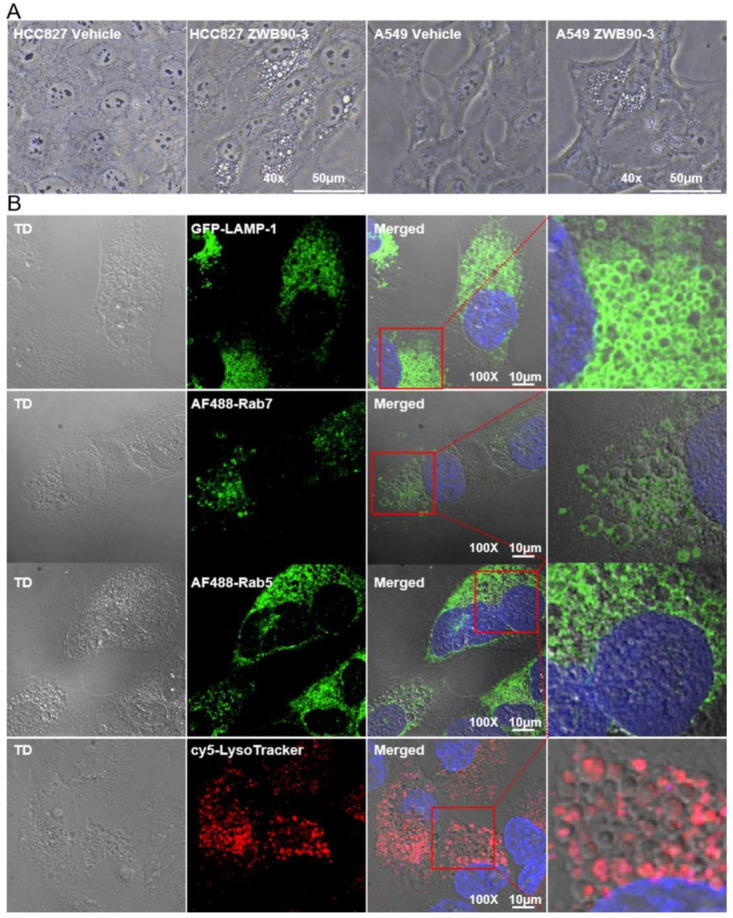
ZWB90-3-induced large vacuole formation in HCC827 and A549 cells. (**A**) Phase contrast images were taken after 72 h of ZWB90-3 or PBS treatment (vehicle) in HCC827 and A549 cells. Images were acquired at 40× magnification. Scale bars: 50 μm. (**B**) HCC827 cells were treated with ZWB90-3 to induce vacuole formation. Confocal imaging was used to co-localize GFP-LAMP1, AF488-Rab7, AF488-Rab5, and cy5-LysoTracker. Images were acquired at 100× magnification. Scale bars: 10 μm.

**Figure 6 ijms-23-12018-f006:**
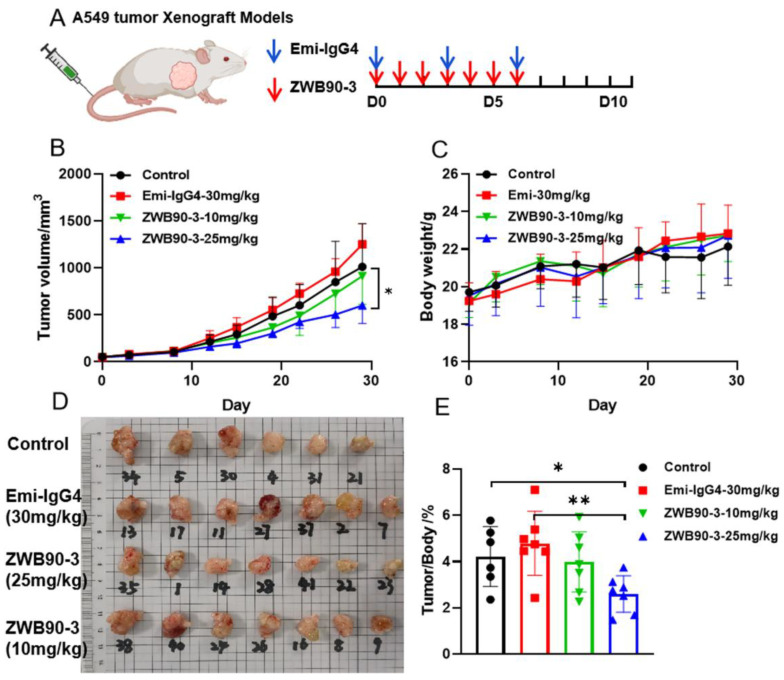
Anti-tumor effects of ZWB90-3 in the A549 NSCLC xenograft mouse model. (**A**) Drug administration scheme. (**B**) Mice were intravenously injected with 10 mg/kg and 25 mg/kg doses of ZWB90-3, and the two control groups were treated with PBS or 30 mg/kg of Emi-IgG4. A total of seven mice per group was used for the study. * *p* < 0.05 was considered significant as compared to the vehicle control. (**C**) Mice body weight curve. (**D**) Tumor tissues were dissected and photographed. (**E**) Tumors were weighed and the tumor-to-body weight ratio was calculated. * *p* < 0.05 was considered significant as compared to the vehicle control. ** *p* < 0.01 was considered significant as compared to Emi-IgG4 group.

## Data Availability

All data supporting the findings of this study are available within the article and its Appendix A.

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
