# Peer review of "Anti-c-MET Fab-Grb2-Gab1 Fusion Protein-Mediated Interference of c-MET Signaling Pathway Induces Methuosis in Tumor Cells"

_ijms, 2022, doi:10.3390/ijms231912018_

Round 1
Reviewer 1 Report
Overview: The authors describe a process for targeting cell proliferation biomarkers (MET/Grb1/Grb2 complex) using a chimeric Fab formatted anti-c-MET antibody (Emibetuzumab) fused to an anti-Grb1/Grb2 peptide (ZWB90-3). Internalization of the Fab is accomplished upon binding to extracellular cMET domain. Their chimeric Fab: downregulated phosphorylation of Gab1, showing tumor suppression; hindered the formation of microtubules via vacuole formation.
The authors provide adequate evidence of each of their central claims (e.g. Enhanced endosomal escape is supported by internalization rate of Emi-Fab-GFPb11 vs Emi-IgG4-GFPb11).
Final paragraph of section 2.1 (line 115-119) needs further explanation (e.g. the significance of binding Trim21).
Explicitly showing the complete amino acid sequence of custom protein constructs would help to ensure reproducibility of the study. The amino acid sequences shown in Supplementary Table S1 are useful in this way; however, complete amino acid sequences of the all other custom constructs (e.g. Emi-Fab-Gab1, Emi-Fab-GFPb11) should be added to the supplement as well.
Spelling: lines 96, 183
Clarity of writing: lines 227-229, 274
Author Response
Dear Reviewer:
Thank you for your letter and for the reviewers’ comments concerning our manuscript entitled “Anti-c-MET Fab-Grb2-Gab1 fusion protein-mediated interference of c-MET signaling pathway induces methuosis in tumor cells” (ID: ijms-1902319). Those comments are all valuable and very helpful for revising and improving our paper, as well as the important guiding significance to our research. We have studied comments carefully and have made the correction which we hope meet with approval.
The revised portion is marked in red in the paper. The main corrections in the paper and the responses to the reviewer’s comments are as flowing:
Responds to the reviewer’s comments:
1.Response to comment: The significance of binding Trim21(Final paragraph of section 2.1).
Response: TRIM21 is a member of the tripartite motif (TRIM) protein family of RING E3 ubiquitin ligases. The TRIM21 interaction with immunoglobulins was found to occur with specificity and extremely high affinity, binding to residues on Fc domain from both CH2 and CH3. The TRIM21/Fc interaction mediated proteasomal degradation. So we speculate that Emi-Fab could avoid interaction with TRIM21 to reduce protein degradation.This will, however, more validation studies need to be performed. About paper,we have re-written this part to introduce more clearly and moved this part from the results to the discussion.
2. Response to comment: Complete amino acid sequences of the all other custom constructsshould be shown.
Response: Thank you for your nice comments on our article. According to your suggestions, we have supplemented complete amino acid sequences of the all other custom constructs in the Supplementary Table S1.
We tried our best to improve the manuscript and made some changes in the manuscript. These changes will not influence the content and framework of the paper. And here we did not list the changes but marked in red in revised paper.
We appreciate for your warm work earnestly and hope that the correction will meet with approval.
Once again, thank you very much for your comments and suggestions.
Thank you and best regards.
Yours sincerely,
Xiaoqian Dou
E-mail : dxiaoqian1991@126.com

Reviewer 2 Report
The authors present data on c-MET targeted Fab-fusion proteins that disrupt the downstream signaling cascades of cMET mediated signaling. The authors demonstrate that monovalent Fab fragment of antagonistic monoclonal antibody Emibetuzumab, has more efficient endosomal escape to the cytoplasm when compared the IgG4 version. The authors use the Fab fragment of Emituzumab as a vehicle to deliver protein domains/eptiopes that are hypothesized to disrupt recruitment of Gab1 and Grb2 to phosphorylated cMET. Fusion protein ZBW90-3 showed inhibition of cMET mediated intracellular signaling, inhibited cell growth and induced vacuole formation in cancer cell lines in vitro and slowed tumor growth in vivo.
The findings in the paper are interesting and the data presented in the manuscript is sound. However, several questions need to be answered.
Comments:
1. The ideal control for ZBW90-3, ZBW90-1 would be the Emi-Fab fragment and not Emi-IgG4. It is disappointing that the authors use Emi-IgG4 as the control throughout the study. This is important for the following reasons:
i) Agonistic activity of monovalent Fab fragment can be better than the bivalent IgG4 version. The authors acknowledge this in lines 172-174, Page 6.
“Emi-IgG4 upregulation effect might due to bivalent Emi-IgG4-mediated dimerization and autophosphorylation of c-MET.Our data confirmed previous reports that monovalent one-arm antibody was an effective tool to against c-MET[31,32]”
ii) In this context, it is difficult to ascertain whether the major contribution to antagonistic activity comes from the monovalency or the dual intervention mechanism described in the manuscript. The fact that ZBW90-3 had better antagonistic activity compared to ZBW90-1 does support the findings that Grb2-SH2 domain in the fusion protein is blocking intracellular signaling cascades. However, it is not clear if the Gab1-MBD is functional in ZBW90-1. Without a clear comparison with monovalent Emi-Fab, it is difficult to support the functionality of Gab1 epitope used in this study.
iii) The authors state in Lines 300-320 “This proved that the part of ZWB90-3 targeting intracellular targets interfered with the intracellular signaling pathway, and could downregulate the phosphorylation of Gab1 and downstream signaling molecules from Gab1”.
It is possible that ZBW90-3 can sterically block the Gab1 binding site, through the binding of Grb2-SH2 alone. Without comparison of Emi-Fab and ZBW90-1, it is difficult to conclude that Gab1-MBD is functional in blocking intracellular signaling.
2. While the western blots do show an overall trend in decreased phosphorylation of cMET, Gab1, AKT, SHP2. The variation is statistically significant only in the case of pAKT. Only a modest increase in RAS concentration is seen and the authors do not indicate the statistical significance of this finding. Therefore, it is not appropriate to use strong/accentuating language as follows:
i) Lines 167-169 “ZWB90-3 treatment caused remarkably reduced in phospho-c-MET level, accompanying by decreased phosphorylation of Gab1, AKT and SHP2 level compared with Emi-IgG4 treatment.”
ii) Lines 318-320, “Similarly, in our study, ZWB90-3 induced the upregulation of intracellular Ras. As a result, ZWB90-3 affected endosome-lysosome fusion and inhibited acidification as well as protein degradation in the lysosomes of cultured cells”. Given that this is an important argument, why is the data in supplementary information?
3. The manuscript needs significant editing before publication to improve readability, correct grammatical errors (several instances throughout the manuscript) and provide missing details. I found the manuscript very difficult to follow, especially the methodology used when describing the results section. Some examples are illustrated below:
I) Lines 90-92 “Consistent with pre-90 vious reports, Emi-IgG4 was observed to cause substantial depletion of c-MET on the cell surface. While Emi-Fab exhibited only partial depletion of c-MET at the same molar concentrations(Figure 1A).”
It is not clear what methodology was used for Figure 1A. I am assuming it is normalized MFI from flow cytometry experiments
iii) Lines 93-95 “Moreover, confocal microscopy showed Emi-Fab induced antibody internalization and displayed a similar intracellular distribution compared to Emi-94 IgG4 in the HCC827 cells (Figure 1B).”
How was the internalization visualized/stained? The results section is difficult to follow without more detail. Moreover, the antibody used for detecting internalization is missing from the methods and materials
Overall, the manuscript describes an important finding which can potentially translate into a vehicle for intracellular delivery of peptide/protein inhibitors. However, as described above additional control experiments and major revisions are necessary. I recommend that the manuscript be accepted pending a major revision.
Author Response
Dear Reviewer:
Thank you for your nice comments on our article titled “Anti-c-MET Fab-Grb2-Gab1 fusion protein-mediated interference of c-MET signaling pathway induces methuosis in tumor cells” (ID: ijms-1902319). Those comments are valuable and shed light on our research. We studied the comments carefully and made corrections which we hope to meet approval.
The revised portion is marketd in red in the paper. Please find attached the main corrections in the paper and the responses to your comments.
We tried our best to improve the manuscript and made some changes in the manuscript. MedEditing have further polished the language of this manuscript. These changes will not influence the content and framework of the paper.
And here we did not list the changes but marked in red in revised paper.
We appreciate for your warm work earnestly and hope that the correction will meet with approval.
Once again, thank you very much for your comments and suggestions. These comments are valuable and helpful for improving the article.
Thank you and best regards.
Yours sincerely,
Xiaoqian Dou
E-mail : dxiaoqian1991@126.com

Round 2
Reviewer 2 Report
The authors have addressed review comments convincingly and provided additional data to support the conclusions drawn in the manuscript. It would strengthen the manuscript, if this additional data is added to supplementary information and discussed briefly in the manuscript.